# RIAM: A Universal Accessible Protocol for the Isolation of High Purity DNA from Various Soils and Other Humic Substances

**DOI:** 10.3390/mps5060099

**Published:** 2022-12-16

**Authors:** Alexander G. Pinaev, Arina A. Kichko, Tatiana S. Aksenova, Vera I. Safronova, Elena V. Kozhenkova, Evgeny E. Andronov

**Affiliations:** 1All-Russia Research Institute for Agricultural Microbiology, Podbelskogo sh. 3, Pushkin, 196608 Saint-Petersburg, Russia; 2Biological Faculty Department of Microbiology, Saint Petersburg State University, Universitetskaya Embankment, 7/9, 199034 St. Petersburg, Russia; 3Dokuchaev Soil Science Institute, 119017 Moscow, Russia

**Keywords:** soil DNA isolation, high phosphate, CTAB DNA precipitation

## Abstract

A single universal open protocol RIAM (named after Research Institute for Agricultural Microbiology) for the isolation of high purity DNA from different types of soils and other substrates (high and low in humic, clay content, organic fertilizer, etc.) is proposed. The main features of the RIAM protocol are the absence of the sorption–desorption stage on silica columns, the use of high concentrations of phosphate in buffers, which prevents DNA sorption on minerals, and DNA precipitation using CTAB. The performance of RIAM was compared with a reference commercial kit and showed very good results in relation to the purity and quantity of DNA, as well as the absence of inhibitory activity on PCR. In all cases, the RIAM ensured the isolation of DNA in quantities much greater than the commercial kit without the effect of PCR inhibition up to 50 ng DNA per reaction in a volume of 15 µL. The latter circumstance along with the ability of the protocol to extract low molecular weight DNA fractions makes the method especially suitable for those cases where quantitative assessments, detection of minor components of soil microbiota, and completeness of isolation of all DNA fractions are required.

## 1. Introduction

For many years, the isolation of DNA from the soil has been the one of the most difficult tasks compared to DNA isolation from other substances. The main reasons are the difficulties in getting rid of humic substances of the soil, the presence of which, even in small amounts, can completely inhibit PCR and DNA sorption on minerals, which significantly reduces the yield of DNA. Most of these issues have been resolved in the commercial kits, but their disadvantages are high prices, closed protocols that limit the use of commercial kits in long-term projects, and the widespread use of the sorption–desorption stage on silica columns, which can also create a number of problems (column shelf life, loss of low molecular weight fractions of DNA, and reduced yield). 

To date, a fairly large number of methods for extracting DNA from the soil including commercial kits and open laboratory protocols have been proposed and continue to be developed [1,2,3,4]. Most of the developed methods focused on solving one of the problems, such as high clay or humic substances content. The reason that inclined us to develop a new protocol for isolating total DNA from soil was to overcome most of the difficulties in a single universal protocol that provides both high DNA yield and high purity. Of particular concern was the avoidance of DNA purification by sorption–desorption on silica columns, which often leads to the significant decrease in the yield of DNA, as well as to the loss of low-molecular fractions of soil DNA, analysis of which may be important for the studies requiring the isolation of the entire array of soil DNAs, belonging to living microorganisms and the extracellular pool of DNA (eDNA), both high and low molecular weight fractions. 

The problem of DNA/RNA sorption on clay soil components has been known for a long time. It is believed that sorption happens mainly due to the cation-mediated interaction of negatively charged phosphate residues of the polysaccharide skeleton of nucleic acid molecules with a negatively charged surface of clays, which are complex aluminosilicate minerals [5], while polyvalent cations are much more effective in the formation of DNA–clay complexes than monovalent ones [6,7]. Known methods for solving this problem are based, firstly, on the use of blocking agents of organic or inorganic origin (ATP, proteins, deoxynucleotide triphosphates, lactose, Ficoll, yeast RNA, PVP, Skim Milk, and modified nucleotides), which compete with DNA molecules for binding centers on clay particles, and secondly, on the selective removal of polyvalent cations (Ca^2+^, Mg^2+^, Zn^2+^, Fe^3+^, and Al^3+^) from soil suspensions using chelating agents such as EDTA or EGTA [4,8,9,10,11]. Biological materials used to block DNA binding centers have a single, but a very significant, drawback: they can contain fragments of nucleic acids of the organisms from which they were obtained, which is especially critical in metagenomic studies.

The simplest and most effective blocker of DNA binding sites on clay particles is the PO_4_^3−^ phosphate ion [8,12,13,14]. However, the concentration of phosphates in the solution, especially when extracting DNA from clay soils, must be quite high up to 1 M in order to maintain the possibility of working in small volumes. The use of phosphate as a blocker has one inconvenience: at phosphate concentrations above 200 mM, the standard protocol for precipitating DNA with ethanol or isopropanol becomes impossible, since, at such concentrations, phosphates are precipitated together with DNA in the form of a drop immiscible with alcohol at the bottom of the tube [15]. The most obvious way to solve this problem is to dilute the DNA preparation by 5–10 times before precipitation with alcohol [1,13], but in this case, the mass isolation of DNA from the soil becomes inconvenient. In this RIAM protocol we found a way to use high phosphate concentration, which not only does not interfere with DNA precipitation, but also helps to get rid of humic components in the next step. 

In the RIAM protocol, we unconventionally use the cationic detergent CTAB to *precipitate* DNA from solutions with high concentrations of phosphates. At NaCl concentrations not higher than 600 mM, CTAB forms a complex with DNA molecules, which is easily precipitated by centrifugation, and at higher salt concentrations, the CTAB-DNA complex is destroyed, and the components of the complex go into solution, where they can easily be separated by extraction with a mixture of chloroform and isoamyl alcohol [16]. CTAB has long been used in the isolation of DNA from the soil [3,17,18,19]; however, its function has always been to form complexes with proteins, polysaccharides, and humic components of the soil insoluble in lysis buffers with a high concentration of NaCl. In our case, the formation of the DNA-CTAB complex under conditions of low NaCl concentration not only allows DNA to be precipitated by centrifugation, but also stabilizes DNA molecules, making them less susceptible to various damaging agents [20].

The main requirement for DNA preparations isolated from the soil is the absence of humic components of the soil (humic acids and fulvic acids), which, even in very small quantities, block the work of polymerases during PCR. Chromatographic methods are most often used to purify crude DNA preparations from humic components, such as the sorption of DNA on silica, followed by washing and elution of bound DNA. This method is used in most commercial soil DNA extraction kits. Less common are the methods based on the sorption of humic components on columns with PVPP [21], gel size exclusion chromatography of DNA preparations on large-pore matrices [22,23,24], purification of DNA from humic components using agarose gel electrophoresis [25], and using activated charcoal [26]. Other methods are based on the selective formation of insoluble complexes with humic components that are removed from solutions by simple centrifugation. The most commonly used technique is DNA precipitation with polyethylene glycol (PEG) with a molecular weight of 6000 or 8000 in the presence of various concentrations of NaCl [1,27]. Inorganic salts, Al(NH_4_)(SO_4_)_2_ and CaCl_2_, more precisely, their cations [28,29], form insoluble complexes with various soil components, including humic ones, so they are often used to wash soil samples until bacterial cells are destroyed or in the process of destruction. In the RIAM protocol we used CaCl_2_ salt for the removal of soil humic substances from DNA preparations, which is added to a lysing solution containing a high concentration of phosphates (1 M). All introduced Ca^2+^ ions form insoluble salts with phosphates, which, however, retain the ability to bind the humic substances of the soil [30] forming poorly soluble Ca-phosphate-humic complexes. Since there are no free Ca^2+^ ions in the solution, there is no danger of nucleic acid precipitation or their interaction with the clay components of the soil.

Thus, in the present study, we have developed a new protocol RIAM that allows us to cope very effectively with the described difficulties, for various types of soils, specifically with high content of humic substances and clay. The hallmark of the protocol is the use of high concentrations of phosphate, the precipitation of humic substances with a calcium containing solution, and the unconventional use of CTAB for DNA precipitation.

## 2. Experimental Design

Three contrast soil types were used for DNA isolation: chernozem (high humic), sod-podzolic soil 1 (low clay and low organic), sod-podzolic soil 2 (high clay and low organic), and organo-mineral fertilizer OMUG produced from poultry dung with a very high content of humic substances [31]. As a reference kit NucleoSpin Soil (Macherey-Nagel, Düren, Germany) was used. This kit was chosen for two reasons. Firstly, we have been successfully using this kit for several years and were completely satisfied with the convenience of working with it and the quality of the isolated DNA. Secondly, this kit gives the opportunity to use several buffer options suitable for soils with different humic and clay contents. 

For each DNA isolation variant, 0.2 g soil in three replications was used. Four buffers suggested by the NucleoSpin Soil kit were used: SL1, SL2 (700 µL) each with and without added Enhancer SX (150 µL). According to the manual, Buffer SL1, in combination with Enhancer SX, is more suitable for soils consisting predominantly of minerals while Buffer SL2 is more suitable for soils with a high amount of organic carbon; however, the best combination can only be determined experimentally. Enhancer SX ensures the highest possible DNA yield but reduces the purity of DNA. The formulas of kit buffers are unknown. In this study, we used all four possible combinations. Finally, isolated DNA samples were dissolved in 50 µL SE buffer from NucleoSpin Soil kit. 

Five microliters of each DNA sample was loaded in 1% agarose gel and separated for 1 h by 5 v/cm in 0.5 × buffer TAE and documented with a digital camera. The exact amount of DNA was measured fluorometrically with Quibt4 (Thermo Fisher Scientific, Waltham, MA, USA). A260/280 and A260/230 were estimated with NanoPhotometr NP80 (Implen Inc., Westlake Village, CA, USA). The PCR inhibition effect was tested with standard PCR protocol with primers targeting the fragment of the variable region V4 of the 16S rRNA gene using universal primers (515F-GTGCCAGCMGCCGCGGTAA/806R-GGACTACVSGGGTATCTAAT) [32]. For analysis of the taxonomic structure of soil microbiome, the deep sequencing of the above-mentioned amplicon library of 16S rRNA gene fragment was performed. The library preparation using primers 515F/806R together with linkers and unique barcodes was performed on a T100 Thermal Cycler (BIO-RAD Laboratories, Heracles, CA, USA) in 15 µL of a reaction mixture containing 0.5 units of Q5^®^ High-Fidelity DNA Polymerase (New England BioLabs, Ipswich, MA, USA), 1X Q5 Reaction Buffer, 5 pM of each primer, 2 mM dNTP (LifeTechnologies, Carlsbad, CA, USA), and 1–5 ng DNA template. The PCR program included the stage of denaturation at 94 °C—1 min, amplification of the product for 25 cycles (94 °C—30 s, 50 °C—30 s, 72 °C—30 s), and final elongation at 72 °C—3 min. Further sample preparation and sequencing were carried out in accordance with Illumina protocol (“16S Metagenomic Sequencing Library Preparation”) on the Illumina MiSeq platform (Illumina Inc., San Diego, CA, USA) using the MiSeq Reagent Kit v3 (600 cycles) with pair-end reading (2 × 300 b) (Illumina Inc., San Diego, CA, USA).

The initial data processing, including demultiplexing and adapter trimming, was performed using Illumina Software (Illumina Inc., San Diego, CA, USA). For further denoising, merging reads, inferring amplicon sequence variants (ASV), and removing chimera, the dada2 [33] package in R software was used. For the taxonomic classification of ASVs, the DECIPHER [34] package was used. To train the classifier, we extracted v4 rRNA fragments from the SILVA database (138 release) [35] records and used them as a training set for the function learnTaxa (DECIPHER). The classification of ASVs was performed with the IdTaxa function with a confidence threshold equal to 70. To construct a phylogenetic tree, the SEPP fragment insertion algorithm implemented in the QIIME2 plugin was used [36]. The QIIME2 package was used to conduct the beta-diversity analysis using PCoA based on the weighted UniFrac metric [37]. The raw sequence data are available in the SRA under the accession number PRJNA892059.

Buffers for RIAM protocol:

Sα

1 M (NH_4_)_2_HPO_4_

100 mM Tris-HCl, pH 8.0

100 mM EDTA

0.5 M Guanidine hydrochloride

Before use add β-Mercaptoethanol to 1% (*v*/*v*)

Sβ

150 mM CaCl_2_

35% (*v*/*v*) Ethanol

Sγ

5% (*w*/*v*) CTAB

3% (*v*/*v*) Triton X100

50 mM EDTA

Sδ

400 mM NaCl

1.5% (*w*/*v*) CTAB

20 mM EDTA

Sε

1 M NaCl

20 mM EDTA

Phenol:(Chloroform/Isoamyl Alcohol) 30:70 saturated with 10 mM Tris, pH 8.0, 

1 mM EDTA

Chloroform:Isoamyl Alcohol 24:1

96% (*v*/*v*) Ethanol

70% (*v*/*v*) Ethanol

5 mM Tris-HCl pH 8.0

Garnet abrasive (GMA Garnet 80 Mesh) 

MOLYKOTE High Vacuum Grease (as a phase-lock media)

All buffers used were sterilized by filtration through 0.22 µm pore size membrane filter.

Those who avoid working with phenol can change this solution for Chloroform:Isoamyl Alcohol 24:1 without loss of DNA quality. However, in this case, there will not be the advantage of using a vacuum grease as a phase-lock media (see Section 3. Procedure step 5).

The RIAM protocol consists of 5 basic stages. 1—disruption of soil microbiota using a homogenizer in solution with a high concentration of phosphate; 2—precipitation of humic substances with Ca^2+^ containing solution; 3—DNA precipitation with CTAB; 4—CTAB elimination with chloroform; 5—DNA precipitation with ethanol. The whole procedure takes about 2.5 h for 24 soil samples.

### 2.1. Important Materials

Only standard laboratory equipment, disposals, and reagents are required. Some important positions are listed below. Instead of these materials and equipment, you can use any analogs. These positions are given here as references.

### 2.2. Materials

GMA 80 Mesh Garnet Abrasive Media (GMA Group, Melbourne, Australia);MOLYKOTE Vacuum grease (DuPont, Wilmington, DE, USA).

Garnet abrasive, used in industries as a blasting agent for cleaning and waterjet cutting, is a perfect and cheap substitution for other lysing matrices such as glass or ceramic beads. High density, which is the main efficiency factor (4.10 Kg/dm^3^), hardness (7.5–8.0 Mohs), chemical resistance, and extremely low price (about $30 for 25 kg) make it a perfect choice compared to glass beads (at least) with lower density (2.5 Kg/dm^3^), hardness (5.5 Mohs), and price (a hundred times more expensive). Fraction 80 Mesh (0.18–0.3 mm) is quite universal for pro- and eukaryotic cell disruption. Before use, rinse several times in tap water, once in distilled water, autoclave for 30 min at 1 atmosphere and dry at 200 °C.

MOLYKOTE Vacuum grease or other silicon-based lubricants with appropriate density can be optionally used in this protocol. It acts like a phase-lock media, forming a tight plug between the aqueous and phenol-chloroform phases, separating problematic interphase from the supernatant, which allows easy pouring of the interphase-free supernatant. Before use autoclave the grease for 30 min at 1 atmosphere. 

### 2.3. Equipment

Precellys Evolution Homogenizer (Bertin Technologies SAS, Montigny-le-Bretonneux, France);Microcentrifuge.

## 3. Procedure

Place a sample of soil (50–300 mg) in a 2 mL test tube with a screw cap, add 300 µL of Sα solution to the tube and suspend the soil by vigorous shaking on a shaker for 5–10 min.Add to the suspension 1 g of Garnet abrasive (fraction 80 Mesh, 0.18–0.3 mm) and 500 µL of a mixture of phenol/chloroform taken in a ratio of 30/70 parts, and tightly tighten the caps;Place the tubes in a bead homogenizer (Precellys Evolution, France) and shake at 6000 rpm 2 times for 30 s;Centrifuge the homogenate at a maximum speed of (16,000–20,000 g) for 2 min.Add 500 μL of Sβ solution to the same tube, close the tube, mix the contents by turning the tube upside down 8–10 times, and centrifuge at maximum speed for 5 min. (To speed up the isolation process at this stage, after adding the Sβ solution, approximately 1 g of silicone grease (Dow Corning High Vacuum Grease, Molykote) can be added to the tube, which after centrifugation forms a tight plug between the aqueous and phenol-chloroform phases, which allows to separate the aqueous phase simply by pouring from test tube to test tube without tedious pipetting. Before use, the lubricant is preferably autoclaved for 30 min at 1 atmosphere);Transfer the aqueous phase into a new 1.5 mL microcentrifuge tube, add an equal volume of Sγ solution (volume determination accuracy ± 10%), mix thoroughly, incubate at room temperature for 5 min and precipitate DNA-CTAB complexes by centrifugation for 5 min at maximum speed. (Volume of the aqueous phase may vary depending on the amount of clay minerals in the soil);Remove the supernatant by inverting the tube, remove the remaining drops by placing the neck of the tube on sterile (autoclaved) filter paper. Complete removal of the supernatant is not necessary. Add 500 μL of Sδ solution to the precipitate, incubate for 5 min at room temperature with vigorous shaking, then precipitate by centrifugation for 5 min at maximum speed;Remove the supernatant as in step 7. Add 400 µL of Sε solution to the pellet and incubate for 5 min at room temperature with vigorous shaking. Then add an equal volume of a mixture of chloroform-isoamyl alcohol (24:1) to the solution and, after vigorous shaking, separate the aqueous and organic phases by centrifugation at maximum speed for 5 min. (For some soils incubation of test tubes at protocol steps 6, 7, and 8 at 65 °C instead room temperature can increase the DNA yield);Transfer the aqueous phase into a new 1.5 mL centrifuge tube, add 2.5 volumes (1 mL) of chilled 96% ethanol, mix well and precipitate the nucleic acids by centrifugation at maximum speed for 10 min;Remove the supernatant, wash the precipitate by adding 400 µL of cold 70% ethanol solution, followed by centrifugation for 5 min at maximum speed;Carefully remove the supernatant (preferably with an aspirator), dry the precipitate for 1 min at 65 °C, and dissolve in the desired volume of autoclaved water or 5 mM Tris-HCl, pH 8.0. To speed up the dissolution, the tubes can be incubated for 5 min at 65 °C.

Figure 1 shows the results of the electrophoretic separation of DNA preparations (5 µL from 50 µL final DNA preparation) obtained using the NucleoSpin Soil kit with four buffer combinations and the RIAM protocol. 

According to the gel electrophoresis data, it can be concluded that the amounts of DNA isolated using the NucleoSpin Soil kit differ significantly depending on the buffer used. However, the RIAM protocol shows a stable result, and the amount of isolated DNA is significantly higher (sometimes five times) than when using the kit (Table 1). Interestingly, in the case of chernozem, the amount of DNA extracted is about 40 µg per g of soil, which is about 1.5 × 10^9^ in terms of the number of *E. coli* genomes, which is close to the maximal expected value and suggests that the RIAM protocol can provide quantitative isolation of DNA from the soil. The purity of the DNA preparation, according to the A 260/280 values, does not differ from the results of the kit; however, according to the A 260/230 values, it significantly exceeds the kit’s results. The latter, apparently, is associated with the residues of guanidine components used in the sorption processes in the kit. There are some differences in the molecular weight of DNAs isolated. In the DNA preparations isolated using the RIAM protocol, the low molecular weight fraction of DNA is highly represented. This may be partly due to differences in the used matrix for the destruction of samples (users can choose a more delicate matrix than garnet abrasive and a less intensive destruction mode), but partly due to the fact that using the RIAM protocol, we isolate all DNA fractions present in the soil, including low molecular weight ones. We are not talking about DNA degradation, since in most DNA samples isolated using RIAM, we can observe a visible band corresponding to 16S rRNA (Figure 1). Most likely, the taxonomic differences revealed between the libraries of 16S rRNA gene corresponding to the kit and the RIAM protocol can be explained precisely by the representation of low molecular weight DNA pools (probably eDNA) with their own specific taxonomic structure (Figure 2, Appendix A). In any case, taxonomic differences due to the use of different kits, including the most popular ones such as MoBio and Macherey-Nagel, are not uncommon [38,39]. The sequence length and GC content distributions in 16S rRNA libraries do not have significant differences between protocols of DNA isolation (Appendix A).

It should be especially noted that the DNA preparation isolated from the highly problematic substrate OMUG differs not only quantitatively, but also in the fact that it does not inhibit the PCR reaction even at a high amount of DNA in the reaction (Figure 3, Table 1). DNA isolated from other soils using the MN and RIAM protocols did not inhibit PCR. The latter is important for the detection of minor components of soil microbiota, since a commonly used dilution of the template DNA before PCR in order to avoid inhibition can lead to the disappearance of minor components of the microbiota from the NGS libraries.

On the whole, it can be concluded that the RIAM protocol makes it possible, using a single universal approach, to isolate DNA from various, including complex, substrates, which is not inferior in both the quantity and purity of the preparation, but in some respects is significantly superior to the commercial kits.

DNA extraction according to the RIAM protocol takes about 2.5 h, this is almost the same as Macherey-Nagel NucleoSpin Soil kit takes. In terms of the cost of isolation, the RIAM method is at least 5–8 times cheaper than the MN kit or MO BIO’s PowerSoil DNA kit. The bulk of the cost comes from plastic disposals and depends on their type.

## 4. Conclusions

A new protocol for DNA extraction from various soils and other humic substances has been developed. The advantages of the RIAM protocol are the possibility of using high concentrations of phosphates without complicating the isolation process; use of CTAB for DNA precipitation, which speeds up and simplifies the purification process; high yield and high purity of DNA preparations isolated from different sources (high and low humic, high and low clay soils and organic fertilizers) and availability of all reagents, minimal use of consumables, and the absence of expensive components. Disadvantages of the RIAM protocol are use of phenol and chloroform, although for those who would like to avoid the use of phenol, it is suggested to use chloroform only. However, in this case, it becomes impossible to use a silicone lubricant as a phase-lock agent. In general, the developed method is a good alternative to commercial kits due to the open protocol, availability of all consumables and reagents, and low cost. The limitation of the study is the small number of studied soils. Since the natural diversity of soils is extremely high, we do not exclude that DNA isolation from some soils using the RIAM protocol may not be as efficient. 

## Figures and Tables

**Figure 1 mps-05-00099-f001:**
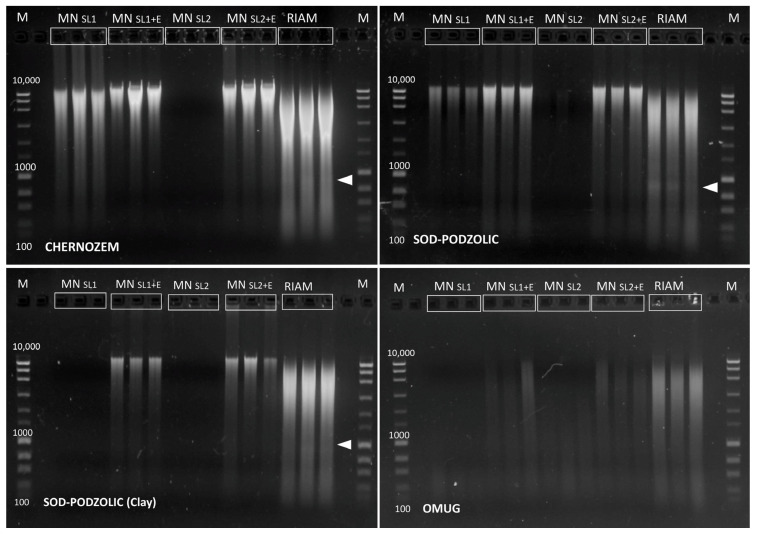
Electrophoretic separation of DNA isolated from four types of substrates with the NucleoSpin Soil kit with four different buffers and with the RIAM protocol developed in this study. M—molecular weight marker. MN SL1, SL1+E, SL2, SL2+E—buffer versions of NucleoSpin Soil kit (Macherey-Nagel, Germany). The white triangle marks a place where a weak band of 16S rRNA is visible.

**Figure 2 mps-05-00099-f002:**
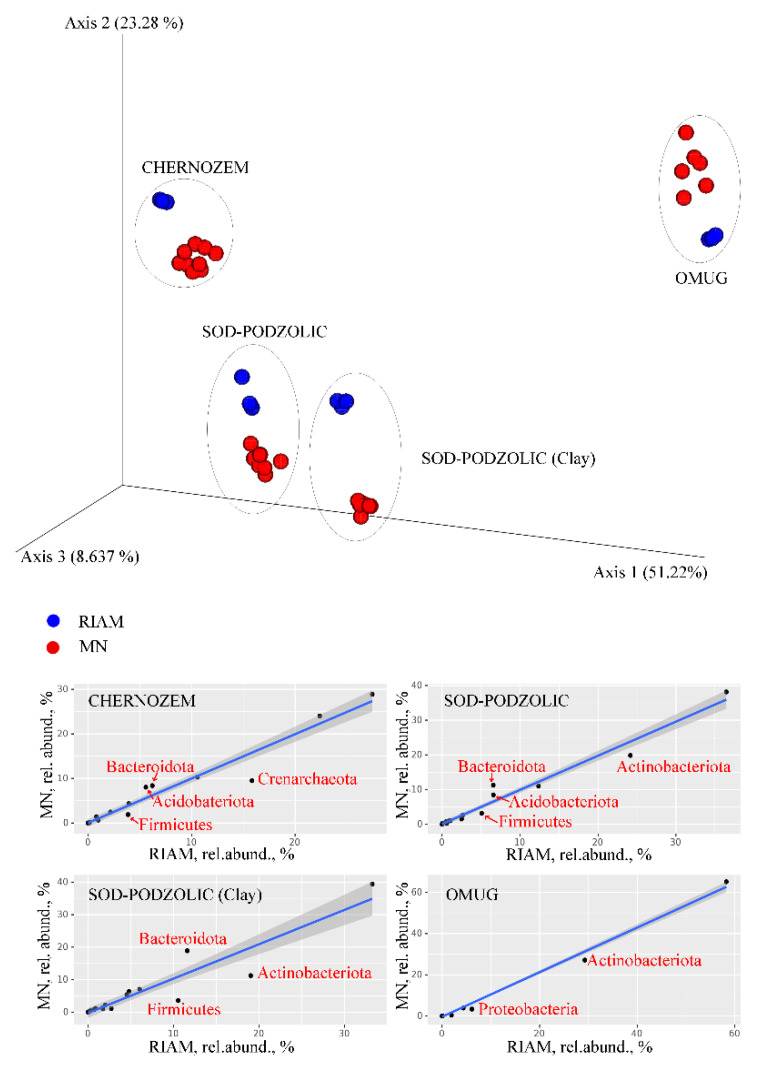
16S rRNA high-throughput sequencing analysis results. In the upper part of the figure, the analysis of β-diversity (PCoA based on the weighted UniFrac metric) is presented. MN—four buffer variants with NucleoSpin Soil kit, RIAM—the protocol developed in this study. In the lower part of the figure comparison of relative abundances of prokaryotic phyla is plotted demonstrating the difference in abundances in MN libraries (the mean for all buffers) and RIAM. Phyla with statistically significant differences in abundance are indicated.

**Figure 3 mps-05-00099-f003:**
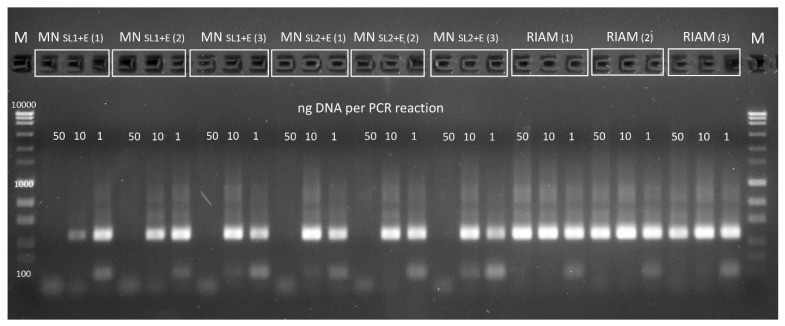
PCR-inhibition effect of 1, 10 and 50 ng DNA isolated with different protocols from OMUG (fertilizer with a high content of humic substances). DNA isolated from other soils using the MN and RIAM protocols did not inhibit PCR.

**Table 1 mps-05-00099-t001:** The concentration of isolated DNAs and purity metrics.

	Soil	MN SL1	MN SL1+E	MN SL2	MN SL2+E	RIAM
conc. ng/µL	chernozem	49.37 ± 5.7	57.93 ± 4.7	nd ^1^	62.90 ± 16.1	168.67 ± 5.7 *
	sod-podzolic	6.25 ± 0.5	25.20 ± 2.0	nd	19.08 ± 3.9	32.73 ± 1.5 *
	sod-podzolic (clay)	nd	8.30 ± 0.9	8.18 ± 1.8	3.43 ± 0.9	42.00 ± 3.5 *
	OMUG	nd	3.34 ± 0.9	nd	3.35 ± 0.4	12.81 ± 1.47 *
A 260/280	chernozem	1.92 ± 0.02	1.91 ± 0.01	nd	1.89 ± 0.01	1.92 ± 0.0
	sod-podzolic	1.68 ± 0.03	1.81 ± 0.06	nd	1.75 ± 0.03	1.86 ± 0.01
	sod-podzolic (clay)	nd	1.87 ± 0.23	2.23 ± 0.82	2.07 ± 0.49	1.91 ± 0.01
	OMUG	nd	1.87 ± 0.26	nd	1.87 ± 0.28	1.87 ± 0.0
A 260/230	chernozem	0.35 ± 0.16	0.33 ± 0.08	nd	0.80 ± 0.22	2.07 ± 0.10
	sod-podzolic	0.34 ± 0.13	0.50 ± 0.30	nd	0.76 ± 0.25	2.07 ± 0.11
	sod-podzolic (clay)	nd	0.34 ± 0.21	0.27 ± 0.15	0.23 ± 0.19	2.07 ± 0.12
	OMUG	nd	0.20 ± 0.17	nd	0.06 ± 0.03	2.07 ± 0.13
PCR inhibition with 50 ng of DNA	OMUG ^2^	nd	yes	nd	yes	no

^1^ for these very low amounts (by electrophoresis data) measurements were not taken. ^2^ The effect is shown only for OMUG, since PCR inhibition is not detected for other soils. * Statistically significant differences in DNA amount between RIAM protocol and the best variant of NucleoSpin Soil kit accordingly to *t*-test (*p* < 0.05).

## Data Availability

The raw sequence data are available in the SRA under the accession number PRJNA892059.

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
