# Peer review of "RIAM: A Universal Accessible Protocol for the Isolation of High Purity DNA from Various Soils and Other Humic Substances"

_mps, 2022, doi:10.3390/mps5060099_

Round 1

Reviewer 1 Report

The authors convincingly presented a new solution to isolate high-purity DNA from a variety of soils and other humic substances that are difficult to purify nucleic acids. The manuscript is well written, and description of all aspects are well presented. Some editing and additional details are required. These are listed below:

1.      Line 139 – Change “MA USA)” to “ MA, USA)”

2.      Line 140 – Change “CA USA)” to “ CA, USA)”

3.      Line 215 – RIAM protocol takes about 2,5 hours for 24 soil samples. How about commercial kits? Add a comment for the run time comparison of the two kits in 3.1. Figures, Tables, and Schemes section.

4.      Line 291 – Change "3.2. Figures, Tables, and Schemes" to “3.1. Figures, Tables, and Schemes”

5.      Line 330 – Add an information about what the soil number is.

6.      Line 340 – Please add a Figure of PCR inhibition experiment with 50ng of DNA.

Author Response

REV 1

Comments and Suggestions for Authors

We thank the reviewer for the kind and substantive comments. We hope that all wishes are taken into account. In addition to the corrections and additions proposed by the reviewers, we have made several minor corrections and additions to the protocol. All changes are marked in red in the text. A supplementary material has been added.

The authors convincingly presented a new solution to isolate high-purity DNA from a variety of soils and other humic substances that are difficult to purify nucleic acids. The manuscript is well written, and description of all aspects are well presented. Some editing and additional details are required. These are listed below:

  1. Line 139 – Change “MA USA)” to “ MA, USA)”

corrected

  1. Line 140 – Change “CA USA)” to “ CA, USA)”

corrected

  1. Line 215 – RIAM protocol takes about 2,5 hours for 24 soil samples. How about commercial kits? Add a comment for the run time comparison of the two kits in 3.1. Figures, Tables, and Schemes section.

added, lines 342-343

  1. Line 291 – Change "3.2. Figures, Tables, and Schemes" to “3.1. Figures, Tables, and Schemes”

changed

  1. Line 330 – Add an information about what the soil number is.

added

  1. Line 340 – Please add a Figure of PCR inhibition experiment with 50ng of DNA.

Figure 2 and text in lines 335-336 added

Reviewer 2 Report

In this study, a new DNA extraction procedure for soils (RIAM) is introduced.  While the results are promising, there are some clarifications required, in my opinion.

In Table 1, the caption should state which soils are referred to be numbers 1, 2, 3 and 4 (i.e. chernozem, sod-podzolic 1, sod-podzolic 2 or OMUG).

I understand that PCR inhibition was tested by PCR with universal primers targeting the variable region V4 of the 16S rRNA gene but what was the criterion for determining whether or not an amplification was inhibited?  I can’t see any relationship between PCR imbibition for soil 4 (yes or no) in Table 1 with the gel images in Figure 1.  Why was PCR inhibition only determined for soil 4 and not soils 1, 2 or 3?

Can the authors please define how beta-diversity was calculated in Figure 2?  The upper part of the figure appears to be a multi-dimensional scaling plot (PCA? PCoA?).  What were the high dimensional data that were reduced to three dimensions in this plot?  What do the numbers within the circles represent?  In the lower plot, are the scales on the horizontal and vertical axes percentages?  If so, please indicate this.

I think that other measures of DNA extraction success should be included in this paper, for example:

·     Raw numbers of on-target (mapped) sequencing reads for each extraction method (MN SL1, MN SL1+E, MN SL2, MN SL2+E and RIAM)

·     Sequencing read length distributions (histograms) for each extraction method (MN SL1, MN SL1+E, MN SL2, MN SL2+E and RIAM)

·     GC content distributions (histograms) for each extraction method (MN SL1, MN SL1+E, MN SL2, MN SL2+E and RIAM)

·     Microbial distributions within each of MN SL1, MN SL1+E, MN SL2, MN SL2+E and RIAM at the level of domain, phylum, class, order, family, genus and species

A cost comparison between DNA extraction using MN and RIAM could be included.

The Conclusions are a collection of bullet points.  This should be expanded as prose and to include as well:

·     Disadvantages of the RIAM protocol (e.g. increased number of steps, use of phenol and chloroform)

·     Any limitations of the study

Author Response

The response is given in a separate file with pictures.

Round 2

Reviewer 2 Report

I think the manuscript is much improved.  The sequence length distributions and GC content distributions could be added to the Supplementary Material, even though they do not differ between MN and RIAM protocols.  They provide further evidence (together with the taxonomy bar graphs) that the RIAM protocol is not significantly changing the DNA representation.  For the sequence length distributions, restrict the horizontal axis range to [200 bp – 300 bp], in 10 bp windows.

Author Response

Dear colleagues,

We have modified figures (scale the sequence lenght distribution to 200-300 n), add this figure together with GC distribution in Supplement (Fig. 6, 7) and add lines 331, 332 in the text. There are also two minor corrections in the text. All modified fragmnents marked with red.

Thanks a lot!